# Attention deficit hyperactivity disorder and educational level in adolescent and adult individuals after anesthesia and abdominal surgery during infancy

Cecilia Arana Håkanson[1]*, Fanny Fredriksson[1,2], Helene Engstrand Lilja[1,2]

1 Department of Women's and Children's Health, Uppsala University, Uppsala, Sweden, 2 Department of Pediatric Surgery, University Children's Hospital, Uppsala, Sweden

* cecilia.arana_hakanson@kbh.uu.se

## Abstract

### Aim

Several studies in animal models have found that exposure to anesthetics in early life can cause cognitive dysfunction. Human studies show conflicting results and studies of cognitive function after anesthesia and neonatal surgery are scarce. The aim of this study was to investigate whether exposure to anesthesia and abdominal surgery during infancy was associated with cognitive dysfunction from the perspective of educational level, disposable income and attention deficit hyperactivity disorders (ADHD) in adolescent and adult individuals.

### Methods

A cohort study with patients born 1976 to 2002 that underwent abdominal surgery during infancy at a pediatric surgical center were matched by age, sex, and gestational age to ten randomly selected individuals from the Swedish Medical Birth Register. Individuals with chromosomal aberrations were excluded. Data on highest level of education and annual disposable income were attained from Statistics Sweden and the diagnosis of ADHD were retrieved from the Swedish National Patient Register.

### Results

485 individuals and 4835 controls were included. Median gestational age was 38 weeks (24–44) and median age at surgery was seven days (0–365). Three hundred sixty-six individuals (70.0%) underwent surgery during the neonatal period (< 44 gestational weeks). Median operating time was 80 minutes (10–430). The mean age at follow-up was 28 years. Fisher's exact test for highest level of education for the exposed and unexposed groups were respectively: university 35% and 33%, upper secondary 44% and 47%, compulsory 21% and 20% (p = 0.6718). The median disposable income was 177.7 versus 180.9 TSEK respectively (p = 0.7532). Exposed individuals had a prevalence of ADHD of 5.2% and unexposed 4.4% (p = 0.4191).

**Data Availability Statement:** Data cannot be shared publicly because of confidentiality. Data are available from the Swedish National Board of

Health and Welfare (contact via
registerservice@socialstyrelsen.se) and Swedish
Ethical Review Authority (contact via
registrator@etikprovning.se) for researchers who
meet the criteria for access to confidential data.

**Funding:** This work was supported by (CAH) HRH
Crown Princess Lovisa's Association for Child
Care, http://www.kronprinsessanlovisa.se/,
grantIDs: 2017-00382; 2018-00455; 2019-00497,
(CAH) the Gillbergska Foundation, http://www.
gillbergska.se/; and (CAH) Specific Clinical
Research Funding from Uppsala County. The
funders had no role in study design, data collection
and analysis, decision to publish, or preparation of
the manuscript.

**Competing interests:** The authors have declared
that no competing interests exist.

## Conclusions

This study shows that exposure to anesthesia and abdominal surgery during infancy is not associated with cognitive dysfunction from the perspective of educational level, disposable income and ADHD in adolescent and adult individuals. Further studies in larger cohorts at earlier gestational ages are needed to verify these findings.

## Introduction

There is increasing concern that exposure to anesthesia and surgery in early childhood may increase the risk of later neurocognitive dysfunction and impaired educational level later in life [1, 2]. Studies in different experimental animal models with mice, rats, pigs and monkeys have found that exposure to commonly used anesthetics during infancy can cause life-long cognitive dysfunction [3–12]. In neonatal piglets, 15 min of surgery designed to replicate an inguinal repair increased cell death in eight areas of the brain compared to anesthesia alone [13].

It is unclear whether these findings in animals can be extrapolated to children. In humans, several studies have reported an increased risk of adverse cognitive development after exposure to general anesthesia and surgery and the risk increased with prolonged or multiple procedures [14–19]. Yet, other studies found that exposure to general anesthesia and surgery in early-life had no association with adverse cognitive development [20, 21]. Possible explanations for the conflicting results have been proposed, such as different indications for surgery and a vulnerability to cognitive dysfunction in infants who undergo anesthesia and surgery [1, 2, 22, 23]. Previous studies have found a doubling of the incidence of learning disabilities (LD) and attention deficit hyperactivity disorders (ADHD) after repeated exposure to general anesthesia before age 2–4 years [17–19]. In a more recent study, an increased risk of ADHD was observed in children under age 5 with a single exposure to minor surgery requiring anesthesia [24].

ADHD is a prevalent neurodevelopmental disorder characterized by symptoms of hyperactivity, impulsivity and inattention [25, 26]. The worldwide prevalence of childhood and adolescent ADHD is estimated to be approximately 5% [27, 28]. Children and adolescents with a diagnosis of ADHD are more likely to drop out of school, to perform poorly on standardized tests and to score lower grades [29].

The aim of this study was to investigate whether exposure to anesthesia and abdominal surgery during infancy was associated with cognitive dysfunction from the perspective of educational level, disposable income and ADHD in adult and adolescent individuals.

## Patients and methods

This study was approved by the ethical review board in Uppsala, Sweden (Dnr 2016/535; Dnr 2016/535/1; Dnr 2016/535/3). Consent was not obtained as the data from the registries were attained and analyzed anonymously.

This cohort study was based on data from the Swedish national population-based registries from a cohort of patients born 1976 to 2002 that underwent anesthesia and abdominal surgery before the age of one year at the Department of Pediatric Surgery at the University Children's Hospital in Uppsala, Sweden. The personal identity number assigned to every Swedish citizen at birth or at immigration was used to link information across registries.

Data from four registers were included in this study: the Swedish Income and Taxation Register (updated annually), with information about disposable income, which is the sum of

all income, excluding taxes, per individual and year, the Education Register which reports the highest level of education achieved by all Swedish individuals from 16 years of age (started 1985 and updated annually), the Swedish Medical Birth Register which includes 98% of all births in Sweden since 1973 and it contains information about gender and gestational week, the Swedish National Patient Register which was started in 1964 and in which all hospitals in Sweden have been included since 1987, the register contains information about gender, age, date of admission and discharge. It also contains diagnoses according to the International Classification of Disease, ICD. During the study period different ICDs were used: ICD-8 from 1969–1986, ICD-9 from 1987–1996 and ICD-10 from 1997 and onwards. The register also keeps codes for surgical procedures according to the Classification of Surgery from 1963–1996, and from 1997 the Classification of Surgical Procedures, KKÅ, has been used. Since 2001 the register covers outpatient visits to a physician, including psychiatric care by both private and public caregivers. In the year 2010 the register had almost 100% coverage for inpatient care whereas the coverage for outpatient care was lower, about 80% [30].

## Study cohort

This study was based on a cohort from our previous study of 898 patients that had undergone laparotomy during infancy (the first year of life) between the years 1976 to 2011 at the Department of Pediatric Surgery at the University Children's Hospital in Uppsala, Sweden [31]. Only individuals exposed to abdominal surgery during infancy were included and no other type of surgery or other procedures requiring anesthesia.

Data were extracted from the patients' medical records. Parameters retrieved were gestational age, sex, date of birth, diagnosis and operating time. Each operation report was reviewed to exclude the risk of faulty registration. Patients with Hirschsprung's disease were operated according to Rehbein's procedure with an initial colostomy followed by an anterior resection of the aganglionic segment. No laparoscopic procedures were included [31].

Out of 898 patients, 523 patients matched the criteria of being aged over16 years at study start. Through linkage with the Swedish Patient Register exclusions were made of individuals with any chromosomal aberrations diagnosed using the ICD8-10, ICD-8 (759.30–759.59), ICD-9 (758A-X), ICD-10 (Q90-Q99). ADHD was defined using ICD9-10 codes relevant for the different subtypes of ADHD, the codes for hyperkinetic disorders: ICD-9 (314J, 314W, 314X), ICD-10 (F90.0A, F90.0B, F90.0C, F90.0X).

## Control cohort

From the Swedish Medical Birth Register, ten controls were drawn at random for each individual in the exposed cohort, matched on sex, age and gestational week. Individuals with the exposure, anesthesia and abdominal surgery before one year of age, were excluded through linkage with the Swedish Patient Register. As for the cohort, exclusion for chromosomal aberrations was made.

## Statistical analysis

Categorical data were presented as frequencies or proportions. Continuous data were presented as either mean with standard deviation or median with range. Highest educational level and ADHD were analyzed using Fischer´s exact test. Since the outpatient register started in 2001, we decided to make two different comparisons of ADHD for the cohort. One for the whole cohort and a subgroup analysis for individuals born from 1995 and later. Further, the highest level of education was analyzed using mixed ordinal regression to look at differences between groups and sex. Incidence rate ratio (IRR) shows the probability of having a higher

level of education. Differences in disposable income were analyzed using the non-parametric Mann-Whitney U-test. In order to be able to take the matching into account a mixed ordinal regression was estimated. Taking specific care by treating a case and its matched controls as a cluster would assist in preventing that the results could be biased by for example unequal number of controls between cases due to missing data. All analyses were performed using R version 3.6.0.

## Results

The study cohort consisted of 523 patients. After exclusion of individuals with chromosomal aberrations, a total number of 485 individuals were included (Fig 1). The unexposed, control group consisted of 4835 individuals. In the control group 15 individuals received no data due to inaccurate personal identification numbers. 472 cases had 10 controls, 12 cases had 9 controls and 1 case had 7 controls (Table 1). Mean age and gestational week of the cases was 28.23 years (5.33 SD) and 37.53 weeks (3.63 SD) respectively. Mean age and gestational week of the control group was 28.34 years (5.24 SD) and 37.54 weeks (3.62).

There were 61.4% males in the cohort (Table 2). Median gestational age was 38 weeks (24–44) and median age at surgery was 7 days (0–365), whereas 366 (70.0%) of the exposed individuals underwent surgery during the neonatal period ($< 44$ gestational weeks). During the study median follow-up 14.7 years (0.0–36.), the median number of surgeries was one (1–13), 26.6% had two surgeries and 16.8% had three or more abdominal surgical procedures.

The most common diagnosis for the study cohort was pyloric stenosis 21.4%, followed by duodenal obstruction 10.3%, diaphragmatic hernia 10.1%, gastroschisis 9.8% and Hirschsprungs disease 9.4% (Table 3). Median operating time was 80 min (10–430).

### Academic performance and disposable income

Level of education was divided into three categories: compulsory school, upper secondary school and university education. Fisher's exact test for highest level of education did not show any significant differences (p-value 0.6718) between exposed and unexposed individuals (Table 4). Mixed ordinal regression for educational level did not show any significant differences for gender (p-value 0.2285) or exposure (p-value 0.4094) (Table 5).

The median annual disposable income did not differ significantly between exposed and unexposed individuals, 177.7 versus 180.9 TSEK respectively (p = 0.7532) (Table 6).

Mixed linear regression on disposable income showed that females in the whole cohort had a significantly lower disposable income compared with males on average -34.75 TSEK CI 95% (-53.7- -15.8) (p<0.001) (Table 7).

### Attention deficit hyperactivity disorders

Fisher's exact test for prevalence of ADHD showed no significant differences between the exposed and unexposed individuals. For the whole cohort, exposed individuals had a prevalence of 5.2% and unexposed 4.4% (p = 0.4191). As for the subgroup, exposed individuals had a prevalence of 8.9% and unexposed 7.6% (p = 0.5781) (Table 8).

## Discussion

This cohort registry study examined the association of exposure to anesthesia and abdominal surgery during infancy with cognitive dysfunction from the perspective of educational level, disposable income and prevalence of ADHD in adult and adolescent individuals. The majority

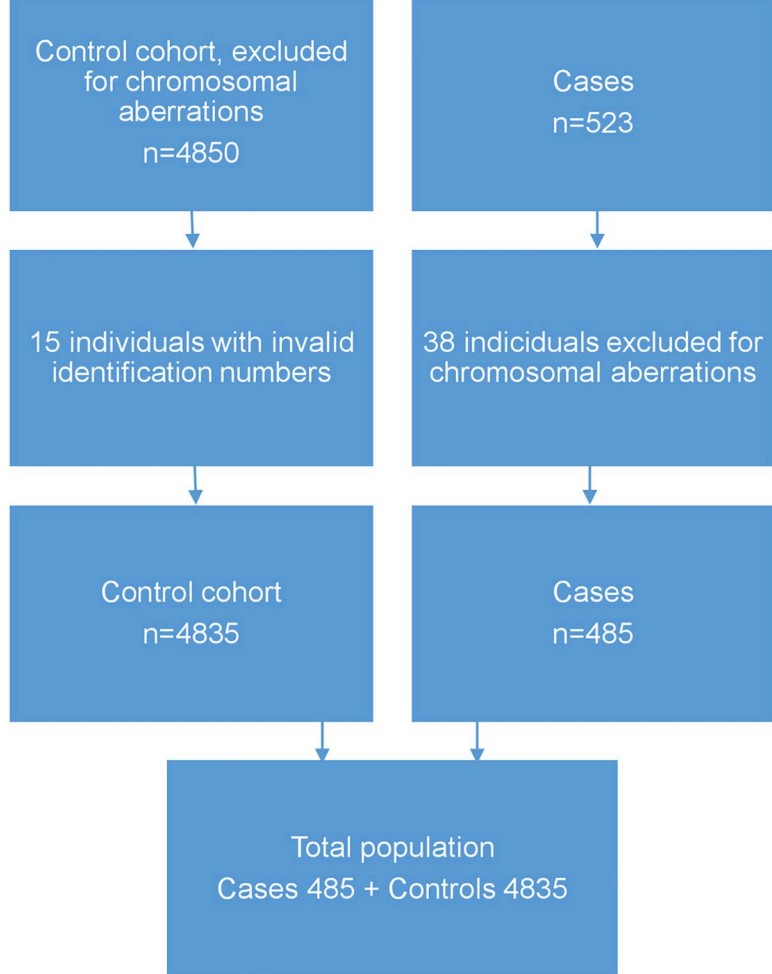

**Fig 1. Flow diagram of the study population.**

of the exposed individuals (70.0%) underwent surgery during the neonatal period ($< 44$ gestational weeks).

We found that the prevalence of ADHD, highest level of education and annual disposable income did not differ significantly between individuals that underwent anesthesia and abdominal surgery during infancy and their matched controls.

In several animal experiments there are strong evidences that exposure to anesthesia in the neonatal period leads to neuronal cell death in the brain and adverse cognitive development [4–6, 8–10, 32, 33] but the results from studies in humans are still conflicting [1, 2, 14–23, 34–36].

**Table 1. Overview of the cohort.**

| Number of controls per case | Number of cases | Number of controls in total | Number of total (cases+controls) |
|---|---|---|---|
| 10 | 472 | 4720 | 5191 |
| 9 | 12 | 108 | 120 |
| 7 | 1 | 7 | 8 |

In the control group 15 individuals received no data due to inaccurate personal identification numbers.

**Table 2. Characteristics of the exposed individuals***.

| Total number of patients | 523 |
|---|---|
| Male gender, n (%) | 321 (61.4) |
| Gestational age in weeks, n** (%) | |
| • <28 | 19 (3.7) |
| • 28–32 | 29 (5.7) |
| • 33–37 | 160 (31.2) |
| • >37 | 305 (59.4) |
| Median gestational age in weeks (range) | 38 (24–44) |
| Median age in days at surgery (range) | 7 (0–365) |
| Median gestational weeks at surgery (range) | 40.1 (25.6–90.1) |
| Median number of surgeries (range) | 1 (1–13) |
| Number of individuals with one surgery (%)*** | 296 (56.6) |
| Number of individuals with two surgeries (%)*** | 139 (26.6) |
| Number of individuals with three or more surgeries (%)*** | 88 (16.8) |

*The cohort of exposed individuals before the exclusion of 38 individuals with chromosomal aberrations.

**Missing data in 10 individuals.

***The total number of surgeries during median follow-up 14.7 years (0.0–36.0).

In contrast to our study, several others have reported an association between exposure to anesthesia and surgery with later LD and poor academic performance [14–16, 18, 19]. A major difference from our study is that they investigated children of pre-school and school age. Our study includes adult and adolescent individuals with a mean age of 28 years. Other possible explanations for the unaffected educational level in the exposed individuals in our study might be the results of adequate educational support with special education and remedial teaching in

**Table 3. Diagnosis and operating time in exposed individuals***.

| Diagnosis | Number of patients, n (%) | Median operating time in min |
|---|---|---|
| | (n = 523) | (range) |
| Pyloric stenosis | 112 (21.4) | 35 (10–110) |
| Duodenal obstruction | 54 (10.3) | 90 (55–220) |
| Diaphragmatic hernia | 53 (10.1) | 85 (25–430) |
| Gastroschisis | 51 (9.8) | 100 (15–200) |
| Hirschsprungs disease | 49 (9.4) | 150 (45–355) |
| Others** | 39 (7.5) | 80 (35–235) |
| Anorectal malformation | 30 (5.7) | 72.5 (30–320) |
| Omphalocele | 27 (5.2) | 92.5 (30–180) |
| Malrotation | 25 (4.8) | 82.5 (55–155) |
| Necrotizing enterocolitis | 19 (3.6) | 82.5 (45–135) |
| Intestinal atresia | 19 (3.6) | 112.5 (65–240) |
| Gastrostomy | 13 (2.5) | 52.5 (35–105) |
| Abdominal tumour | 13 (2.5) | 125 (45–285) |
| Biliary atresia | 11 (2.1) | 200 (55–335) |
| Intussusception | 8 (1.5) | 70 (20–110) |

*Exposed individuals before the exclusion of 38 individuals with chromosomal aberrations.

**Omphaloenteric duct, ovarian cyst, Meckel's diverticulum, intestinal duplication.

**Table 4. Highest educational level.**

|  | Controls | Cases |
|---|---|---|
|  | n = 4221 | n = 454 |
| University | 1391 (33%) | 157 (35%) |
| Upper secondary school | 1967 (47%) | 202 (44%) |
| Compulsory school | 863 (20%) | 95 (21%) |

Fisher's exact test for highest level of education did not show any significant differences between exposed and unexposed individuals (p = 0.6718).

our patients. During normal brain development, 50–70% of the entire neuronal cell population is removed by physiological apoptosis but the brain has a significant capacity to recovery function during childhood [37]. We speculate that this ability of the brain may contribute to the unaffected level of academic performance in the exposed adult and adolescent individuals in our study.

Most previous studies include children exposed to anesthesia and surgery beyond infancy with various diagnoses, different operating times, outcome measures, experimental design and small sample sizes of neonates and infants, making it problematic to compare the results [14–20]. A study including similar diagnoses to the current study found that in early adolescence, children who had undergone neonatal surgery performed less well academically compared with their peers. However, it is hard to draw any conclusions from that study as only 30 patients were included, and it involved 12 different diagnosis [38]. Bartels et al. showed that 1143 monozygotic twin pairs exposed to anesthesia and surgery before the age of three years had at the age of 12 years significantly lower educational achievement scores and more cognitive problems than twins not exposed to anesthesia [20]. However, the unexposed co-twin from discordant pairs did not differ from their exposed co-twin. The authors' conclusion of the study was that there was no evidence for a causal relationship between anesthesia and later LD. Instead, early anesthesia was a marker of an individual's vulnerability to later LD. A retrospective cohort study by Wilder et al. of various types of surgical procedures in children before four years of age found that exposure to anesthesia and surgery was a significant risk factor for the later development of LD in children receiving multiple, but not single anesthetics [19]. However, these results may be explained by confounders as children with more serious diagnosis and comorbidities are more likely to need surgery.

Epidemiologic studies are few but those available are in agreement with our findings [21, 39]. A Danish nationwide study by Hansen et al found no evidence that a single, anesthetic exposure in conjunction with hernia repair in infancy reduced academic performance at age 15 or 16 years after adjusting for known confounding factors [21]. However, the study is difficult to compare to ours as the anesthesia/ operating time was shorter than in our study. A more similar recent Swedish nationwide study found that exposure to anesthesia and surgery before the age of four years had a small association with later academic performance or

**Table 5. Mixed ordinal regression for educational level.**

| Variable | IRR | 95% CI | P-value |
|---|---|---|---|
| Case: yes | 1.09 | (0.89–1.34) | 0.4094 |
| Gender: female | 1.25 | (0.87–1.81) | 0.2285 |

Mixed ordinal regression for educational level did not show any significant differences for gender or exposure.

**Table 6. Disposable income.**

| Income (1000 SEK) | Controls n = 4493 | Cases = 483 |
|---|---|---|
| Median | 177.7 | 180.9 |

Mann-Whitney U-test for disposable income did not differ significantly between exposed and unexposed individuals (p-value = 0.7532).

cognitive performance in adolescence on a population level [39]. In the exposed group, 8.7% of the individuals were born preterm compared to 40.6% in our study. They found no reduction in school grades among children with surgery before the age of one.

The median annual disposable income did not differ significantly between exposed and unexposed individuals in our study, which is supported by a Swedish nationwide population-based cohort study of 389 patients with Hirschsprung's disease [40]. Neither the individual disposable income nor the highest educational level differed between patients with Hirschsprung's disease and controls [40]. Another Swedish registry study including 522,310 individuals born in 1973–1979 reported that preterm birth was associated with a lower chance of completing a university education and a lower net salary [41]. With the matching of controls for gestational week this potential confounder was omitted in our study. To further decrease the risk of confounders, we also matched for age and gender and excluded individuals with chromosomal aberrations. There may be other important differences between the groups that were not measured such as socioeconomic status of the families or baseline health characteristics. The matching could have been made with more potential confounders but it would decrease the number of individuals in the control group. Matching by gestational age had higher priority in our study. We found that exposed as well as unexposed females had significantly lower disposable incomes compared to males. The gender wage gap is well known and has been described in previous studies [42, 43].

The prevalence of ADHD was not significantly increased in exposed individuals in our study. A retrospective study by Sprung et al. including individuals that underwent surgery before the age of two could not find an increased level of ADHD at age 19 following one surgical procedure [17]. However, with repeated exposures to general anesthesia there was an increased risk of development of ADHD later in life. A more recent retrospective study including 573 children exposed to anesthesia and surgery prior to the age of three found that multiple, but not single, exposures were associated with an increased frequency of both LD and ADHD [44]. In recent years there seems to be an increased awareness of ADHD. A Swedish registry study showed that the number of individuals diagnosed with ADHD is increasing. In 2006 the prevalence for the whole population was 1.1 per 1000 persons and in 2011, 4.8 per 1000 persons, the highest increase was seen in females as well as in ages 22 years and above [45]. In 2011, 57.8% of the individuals with an ADHD diagnosis were under the age of 22 [45].

**Table 7. Gender differences for disposable income.**

| Variable | Coefficient | 95% CI | p-value |
|---|---|---|---|
| (intercept) | 201.68 | (190.0–213.4) | <0.001 |
| Case: Yes | 1.74 | (-9.6–13.1) | 0.763 |
| Gender: Female | -34.75 | (-53.7- -15.8) | <0.001 |

Mixed linear regression on disposable income showed that females in the whole cohort had a significantly lower disposable income compared with males.

**Table 8. ADHD diagnosis.**

|  | Controls | Cases |
|---|---|---|
| **All** | **P-value = 0.4191** | |
| No diagnosis | 4623 (95.6%) | 460 (94.8%) |
| Diagnosis | 212 (4.4%) | 25 (5.2%) |
| **1995-** | **P-value = 0.5781** | |
| No diagnosis | 1010 (92.4%) | 102 (91.1%) |
| Diagnosis | 83 (7.6%) | 10 (8.9%) |

Fisher's exact test for prevalence of ADHD showed no significant differences between the exposed and unexposed individuals.

In our study, since the outpatient register started in 2001, we carried out two different analyses for the prevalence of ADHD. We did this since we believed that ADHD would be underestimated for both exposed and unexposed individuals even though the relative difference between the groups was believed to be correct. The subgroup analysis included individuals born in 1995 and later. By doing this we could find an increase of ADHD in both groups but still no significant difference was seen between the groups.

The Swedish National Patient Register is a valuable source for register research, the validity is high but not for all diagnoses [30]. Strengths of the present study were that the diagnoses and exposure were ensured by previous review of the patients' charts and that the matched controls were drawn at random by the National Board of Health and Welfare, thus reducing the risk of selection bias. For the unexposed controls, we used codes for surgical procedures rather than ICD codes for diagnoses to decrease the risk of misclassification. Other strengths are the high number of included neonates and infants that underwent anesthesia and abdominal surgery with a long follow-up time and data on the duration of surgery.

Limitations of this study were that the outcome measure of educational level may not detect subtle effects from anesthesia and surgery in early childhood. Furthermore, various diagnoses among included patients with different operating times may challenge the interpretation of the results. Another limitation to the study was that 15 controls were missing due to invalid identification numbers. However it is unlikely that this would markedly influence the results. As the majority of neonates were exposed at a later gestational age (>33 weeks) the results regarding exposure at much earlier gestational age might be different and can not be generalized.

## Conclusions

This study shows that exposure to anesthesia and abdominal surgery during infancy is not associated with cognitive dysfunction from the perspective of educational level, disposable income and ADHD in adult and adolescent individuals, despite the fact that the majority of surgeries (70%) were in neonates. It is important to identify pediatric surgical diagnoses of high risk of cognitive dysfunction to arrange for early educational support. This knowledge is also important to professionals and caregivers of these patients. Although our results are reassuring, they cannot exclude more subtle effects in cognitive function. Future studies with larger cohorts of individuals that have undergone surgery at earlier gestational ages are needed.

## Supporting information

**S1 File. Excluding codes for abdominal surgery.**
(DOCX)

**S2 File. Overview of the cohort.**
(DOCX)

## Acknowledgments

We would like to express our gratitude to Fabian Söderdahl, Statisticon AB, for his assistance with the statistical analyses and Sheila Macdonald-Rannström for the final language review.

## Author Contributions

**Conceptualization:** Cecilia Arana Håkanson, Fanny Fredriksson, Helene Engstrand Lilja.

**Data curation:** Cecilia Arana Håkanson, Fanny Fredriksson.

**Formal analysis:** Cecilia Arana Håkanson, Fanny Fredriksson.

**Investigation:** Cecilia Arana Håkanson.

**Methodology:** Cecilia Arana Håkanson, Helene Engstrand Lilja.

**Supervision:** Helene Engstrand Lilja.

**Writing – original draft:** Cecilia Arana Håkanson.

**Writing – review & editing:** Cecilia Arana Håkanson, Fanny Fredriksson, Helene Engstrand Lilja.

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
