## [Decision Letter · Decision Letter 0]

16 Jul 2020

PONE-D-20-10556

Exposure to anesthesia and abdominal surgery during infancy is not associated with cognitive dysfunction later in life

PLOS ONE

Dear Dr. Arana Håkanson,

Thank you for submitting your manuscript to PLOS ONE. After careful consideration, we feel that it has merit but does not fully meet PLOS ONE’s publication criteria as it currently stands. Therefore, we invite you to submit a revised version of the manuscript that addresses the points raised during the review process.

We look forward to receiving your revised manuscript.

Kind regards,

Antonio Palazón-Bru, PhD

Academic Editor

PLOS ONE

Journal Requirements:

https://jamanetwork.com/journals/jamapediatrics/fullarticle/2580308

In your revision ensure you cite all your sources (including your own works), and quote or rephrase any duplicated text outside the methods section. Further consideration is dependent on these concerns being addressed.

Reviewers' comments:

Reviewer's Responses to Questions

**Comments to the Author**

1. Is the manuscript technically sound, and do the data support the conclusions?

Reviewer #1: No

Reviewer #2: Partly

2. Has the statistical analysis been performed appropriately and rigorously? 

Reviewer #1: Yes

Reviewer #2: I Don't Know

3. Have the authors made all data underlying the findings in their manuscript fully available?

Reviewer #1: No

Reviewer #2: No

4. Is the manuscript presented in an intelligible fashion and written in standard English?

Reviewer #1: Yes

Reviewer #2: Yes

5. Review Comments to the Author

Reviewer #1: This is case control study to evaluate whether exposure to anesthesia and abdominal surgery during infancy was associated with attainment in educational level and income in adult lives. The strengths of the study is that a uniform type of surgery was performed. However, although the study adds information to this literature, the title of the paper is mis-leading as the study is mis-leading and should be revised to properly reflect what was done.

There are a few areas that need further discussion:

1. The study was conducted over a long period of time. Whether changing practice (such as surgical approaches or anesthetic management) might have affected the outcomes was not discussed.

2. Measurement of ADHD - it is not clear whether it was well captured, as likely not all patients with ADHD sought medical consultation.

3. Age of exposure – it appears that the majority of the subjects were exposed to anesthesia and surgery at a later gestational age (>33 weeks). Whether these results can be generalized to those exposure at much earlier gestational age need to be included in the discussion.

4. Similarly, the number of repeated exposures to anesthesia and surgery was also not high. It is unclear whether the authors only measured repeated abdominal surgery, and not other type of surgery.

Reviewer #2: This is an interesting manuscript evaluating a number of different outcomes in children exposed to anesthesia for abdominal surgery.

Specific comments:

Page 4, Line 73: While some studies have found a near doubling of ADHD, this was only found in the children with multiple exposures. In the same study by Sprung et al., the HR for a single exposure was 1.18, which was not statistically significant. In more recent studies published in Anesth Analg. by Ing et al. the HRs for an increased risk of ADHD was found to be between 1.25 and 1.37.

Page 7, Line 131: It would be helpful to know how many total patients were in the Swedish Medical Birth Register and how they were drawn at random based on sex, age, and gestational week. Were they manually selected by one of the researchers or was there a computer program that would randomly select from anywhere in the database?

Page 7, Line 133: Is it possible that the controls had surgery after age 1 year old? Therefore, if anesthetics had an adverse effect in children between ages 1 and 3 and the controls included children exposed at those ages, is it possible that these results may be biased towards a null effect?

Page 7, Line 147: Is there a reason why the 1995 cutoff was chosen?

Page 7, line 148: It would be helpful to explain why mixed models are needed for these regression analyses. This data does not seem to be hierarchical in nature.

Page 8, Line 157: If there were 485 cases and it is 1:10 matching, shouldn’t there be 4850 controls instead of 4835?

Figure 1: This flow diagram would be more informative if it also included the selection of the controls.

Table 1: It may be more informative to show the demographic characteristics of the 485 patients included in the study. I believe the 523 patients includes patients who were excluded due to chromosomal aberrations. If there was missing gestational age data in 10 patients, how was the matching done? Was matching done on only age and sex for those patients? In the children with multiple procedures (2 or 3 or more) were these all procedures done before age 1, or at any time during their life? It would also be helpful to note the demographic characteristics in the matched controls, which should confirm exact matching and give information on whether they had surgery or anesthesia later on in their life.

Table 2: It may be more informative to show the surgical characteristics of the 485 patients included in the study.

Table 3: If there are n=485 exposed children and n=4835 matched controls, why is there only data for n=454 exposed children and n=4221 controls? Is there missing outcome data for the cases and controls?

Table 4: It would be helpful to define IRR.

Table 7: If there are n=485 exposed children and n=4835 matched controls, why is there only data for n=431 exposed children yet there is data for all n=4835 controls? If this is missing diagnosis data, is there a reason why it is only missing in the exposed children and not the matching controls?

Page 14, Line 279: Given that the incidence of ADHD diagnosis has increased over time, it would be interesting to know if the birth years of the exposed and matched controls were similar. If they were born in different eras, it may bias the results.

Page 15, Line 298: While the exposed children and matched controls were matched on age, sex, and gestational week, is it possible that there may be other important differences between the groups that were not measured such as socioeconomic status of the families or baseline health characteristics?

6. PLOS authors have the option to publish the peer review history of their article (what does this mean?). If published, this will include your full peer review and any attached files.

Reviewer #1: No

Reviewer #2: No

---

## [Author Response · Author response to Decision Letter 0]

17 Aug 2020

Dear Academic Editor.

Thank you for your valuable comments regarding the manuscript. We have taken careful consideration of all the comments and the manuscript has been changed according to yours and the reviewers suggestions. We think that the manuscript has improved and hope that it can be accepted for publication in PLOS ONE. Please see our response on the comments listed below. 

Our manuscript meets PLOS ONE's style requirements, including those for file naming. We have rephrased and omitted some minor occurrence of overlapping text with a previous publication and we have made data underlying the findings in the manuscript fully available as supporting information.

Sincerely 

Cecilia Arana Håkanson, corresponding author 

Department of Women's and Children's Health, Uppsala University.

 e-mail: cecilia.arana_hakanson@kbh.uu.se

Response to reviewers

Reviewer #1: This is case control study to evaluate whether exposure to anesthesia and abdominal surgery during infancy was associated with attainment in educational level and income in adult lives. The strengths of the study is that a uniform type of surgery was performed. However, although the study adds information to this literature, the title of the paper is mis-leading as the study is mis-leading and should be revised to properly reflect what was done.

Response: We have changed the title to “Attention deficit hyperactivity disorder and educational level in adolescent and adult individuals after abdominal surgery during infancy” 

There are a few areas that need further discussion:

1. The study was conducted over a long period of time. Whether changing practice (such as surgical approaches or anesthetic management) might have affected the outcomes was not discussed.

Response: The study was conducted over a long period of time and both surgical approaches and anesthetic management have changed over that period. However, our study uses a case-control approach with the individuals being exposed to abdominal surgery during infancy and the controls not. The cases are not compared to each other but only to their controls matched on sex, age and gestational week. Therefore, these changes should not affect the outcome.

2. Measurement of ADHD - it is not clear whether it was well captured, as likely not all patients with ADHD sought medical consultation.

Response: The measurement of ADHD was drawn from the Swedish National Patient Register, hence we do miss individuals that does not seek medical consultation. However, we believe that there should not be a difference in the pattern of seeking medical consultation between the cases and their controls. 

3. Age of exposure – it appears that the majority of the subjects were exposed to anesthesia and surgery at a later gestational age (>33 weeks). Whether these results can be generalized to those exposure at much earlier gestational age need to be included in the discussion.

Response: We have in the discussion page 16, lines 312-314 added “As the majority of neonates were exposed at a later gestational age (>33 weeks) the results regarding exposure at much earlier gestational age might be different and can not be generalized”.

4. Similarly, the number of repeated exposures to anesthesia and surgery was also not high. It is unclear whether the authors only measured repeated abdominal surgery, and not other type of surgery.

Response: Only exposure to abdominal surgery during infancy were included and no other type of surgery or other procedures requiring anaesthesia. This information is added in Study Cohort page 6 ,lines 122-123.

Reviewer #2: This is an interesting manuscript evaluating a number of different outcomes in children exposed to anesthesia for abdominal surgery.

Specific comments:

Page 4, Line 73: While some studies have found a near doubling of ADHD, this was only found in the children with multiple exposures. In the same study by Sprung et al., the HR for a single exposure was 1.18, which was not statistically significant. In more recent studies published in Anesth Analg. by Ing et al. the HRs for an increased risk of ADHD was found to be between 1.25 and 1.37.

Response: Page 4, lines 76-78 we have added repeated to the sentence ”Previous studies have found a doubling of the incidence of learning disabilities (LD) and attention deficit hyperactivity disorders (ADHD) after repeated exposure to general anesthesia before age 2-4 years (1-3). We also added this sentence: ”In a more recent study, an increased risk of ADHD was observed in children under age 5 with a single exposure to minor surgery requiring anesthesia” (4).

Page 7, Line 131: It would be helpful to know how many total patients were in the Swedish Medical Birth Register and how they were drawn at random based on sex, age, and gestational week. Were they manually selected by one of the researchers or was there a computer program that would randomly select from anywhere in the database?

Response: The Swedish Medical Birth Register includes 98% of all births in Sweden since 1973. Register data is protected by strict confidentiality but can be made available for research after a special application following ethical approval as in our study. The data withdrawal and the random selection were made by statisticians employed at the Register service unit using computer programs.

Page 7, Line 133: Is it possible that the controls had surgery after age 1 year old? Therefore, if anesthetics had an adverse effect in children between ages 1 and 3 and the controls included children exposed at those ages, is it possible that these results may be biased towards a null effect?

Response: Yes, there is a possibility that the controls and their cases had surgery and/or anesthetics after the age of one year. However we set the exposure of the cases to abdominal surgery before the age of one. Since the 4835 controls were drawn at random we believe that the group should reflect the common exposure to surgery and/or anesthetics in the Swedish population in general. 

Page 7, Line 147: Is there a reason why the 1995 cutoff was chosen?

Response: The diagnose of ADHD is mainly found in the outpatient registers and this register started in 2001. The diagnose of ADHD is hard to confirm in very young children. With the cutoff 1995 the individuals were 6 years or older and more likely to have the diagnose code for ADHD in the register.

Page 7, line 148: It would be helpful to explain why mixed models are needed for these regression analyses. This data does not seem to be hierarchical in nature.

Response: In order to be able to take the matching into account a mixed ordinal regression was estimated. Taking specific care by treating a case and its matched controls as a cluster would assist in preventing that the results could be biased by for example unequal number of controls between cases due to missing data. We have added this information under statistics pages 7-8, lines 155-159.

Page 8, Line 157: If there were 485 cases and it is 1:10 matching, shouldn’t there be 4850 controls instead of 4835?

Response: A very valid comment. The control group was drawn at random by statisticians at the Register service unit. When recieving the data from the registers the personal identification number was replaced by a serial number due to confidentiality. In the control group 15 individuals had the same serial number due to inaccurate personal identification numbers, therefore no data would be found on these individuals. 12 were matched to different cases and 3 were matched to the same case. Please see added table in supporting information, S2. 

Figure 1: This flow diagram would be more informative if it also included the selection of the controls.

Response: We have changed the flow diagram to include selection of the controls. 

Table 1: It may be more informative to show the demographic characteristics of the 485 patients included in the study. I believe the 523 patients includes patients who were excluded due to chromosomal aberrations. If there was missing gestational age data in 10 patients, how was the matching done? Was matching done on only age and sex for those patients? In the children with multiple procedures (2 or 3 or more) were these all procedures done before age 1, or at any time during their life? It would also be helpful to note the demographic characteristics in the matched controls, which should confirm exact matching and give information on whether they had surgery or anesthesia later on in their life.

Response: Correct, the 523 patients include the excluded patients. We agree that it would be more informative with the demographic data on only the included 485 individuals. Due to confidentiality in the registers we do not have the information on which ones that were excluded and therefore we were not able to only show the demographic characteristics of only the 485 individuals. The missing gestational age in 10 individuals were when we manually collected the data from the local patients charts at the hospital. The matching of controls was made by the information in the Swedish Birth Register connected to the personal identification code of 10 digits that all citizens are given at birth and not from the data that were manually collected. All controls were matched on age, sex and gestational age (week). This study was based on a cohort from our previous study of 898 patients that had undergone laparotomy during the first year of life between the years 1976 tom 2011. 523 patients matched the criteria of being aged over 16 years at study start. The number of surgeries were the total number of surgeries during the follow up period of the previous study with a median follow up of 14.7 years (0.0 – 36.0). We did not collect register data on whether the controls were exposed to surgery and/or anesthesia later in life. In the children with multiple procedures (2 or 3 or more) only the first procedure had to be done before age 1. Most reoperations were done close in time from the primary surgery e.g 70% undergoing laparotomy for small bowel obstruction occurred within 2 years of the primary surgery. 

Table 2: It may be more informative to show the surgical characteristics of the 485 patients included in the study.

Response: We agree that it would be more informative with the surgical characteristics on only the included 485 individuals. Due to confidentiality in the registers we do not have the information on which ones that were excluded and therefore we were not able to only show the data of the 485 individuals. 

Table 3: If there are n=485 exposed children and n=4835 matched controls, why is there only data for n=454 exposed children and n=4221 controls? Is there missing outcome data for the cases and controls?

Response: 

There are 31 cases and 399 controls that miss the outcome of highest educational level. With a case missing this outcome also its controls were excluded in the analysis. Please see added table in supporting information, S2. 

Table 4: It would be helpful to define IRR.

Response: IRR, incidence rate ratio shows the probability of having a higher level of education. Added in the statistics section page 7,lines 153-154.

Table 7: If there are n=485 exposed children and n=4835 matched controls, why is there only data for n=431 exposed children yet there is data for all n=4835 controls? If this is missing diagnosis data, is there a reason why it is only missing in the exposed children and not the matching controls?

Response: Thank you for your careful review, we made a mistake here. The right number of cases is 460 (and not 406) with no ADHD diagnosis and 25 with the diagnosis. We must have mixed up the numbers when editing the table. Now corrected in the Table 7.

Page 14, Line 279: Given that the incidence of ADHD diagnosis has increased over time, it would be interesting to know if the birth years of the exposed and matched controls were similar. If they were born in different eras, it may bias the results.

Response: The matching of age was made by the same year of birth for cases and controls so this would not be a source of bias. 

Page 15, Line 298: While the exposed children and matched controls were matched on age, sex, and gestational week, is it possible that there may be other important differences between the groups that were not measured such as socioeconomic status of the families or baseline health characteristics?

Response: We are aware that there may be other important differences between the groups that were not measured such as socioeconomic status of the families or baseline health characteristics. The matching could have been made with more potential confounders but it would decrease the number of individuals in the control group. Matching by gestational age had higher priority in our study. We have added this section in discussion page 14, lines 272-276.

1. Sprung J, Flick RP, Katusic SK, Colligan RC, Barbaresi WJ, Bojanic K, et al. Attention-deficit/hyperactivity disorder after early exposure to procedures requiring general anesthesia. Mayo Clin Proc. 2012;87(2):120-9.

2. Flick RP, Katusic SK, Colligan RC, Wilder RT, Voigt RG, Olson MD, et al. Cognitive and behavioral outcomes after early exposure to anesthesia and surgery. Pediatrics. 2011;128(5):e1053-61.

3. Wilder RT, Flick RP, Sprung J, Katusic SK, Barbaresi WJ, Mickelson C, et al. Early exposure to anesthesia and learning disabilities in a population-based birth cohort. Anesthesiology. 2009;110(4):796-804.

4. Ing C, Sun M, Olfson M, DiMaggio CJ, Sun LS, Wall MM, et al. Age at Exposure to Surgery and Anesthesia in Children and Association With Mental Disorder Diagnosis. Anesth Analg. 2017;125(6):1988-98.

---

## [Decision Letter · Decision Letter 1]

8 Sep 2020

PONE-D-20-10556R1

Attention deficit hyperactivity disorder and educational level in adolescent and adult individuals after anesthesia and abdominal surgery during infancy

PLOS ONE

Dear Dr. Arana Håkanson,

Thank you for submitting your manuscript to PLOS ONE. After careful consideration, we feel that it has merit but does not fully meet PLOS ONE’s publication criteria as it currently stands. Therefore, we invite you to submit a revised version of the manuscript that addresses the points raised during the review process.

We look forward to receiving your revised manuscript.

Kind regards,

Antonio Palazón-Bru, PhD

Academic Editor

PLOS ONE

Reviewers' comments:

Reviewer's Responses to Questions

**Comments to the Author**

1. If the authors have adequately addressed your comments raised in a previous round of review and you feel that this manuscript is now acceptable for publication, you may indicate that here to bypass the “Comments to the Author” section, enter your conflict of interest statement in the “Confidential to Editor” section, and submit your "Accept" recommendation.

Reviewer #2: (No Response)

2. Is the manuscript technically sound, and do the data support the conclusions?

Reviewer #2: Partly

3. Has the statistical analysis been performed appropriately and rigorously? 

Reviewer #2: Yes

4. Have the authors made all data underlying the findings in their manuscript fully available?

Reviewer #2: No

5. Is the manuscript presented in an intelligible fashion and written in standard English?

Reviewer #2: Yes

6. Review Comments to the Author

Reviewer #2: The authors did a nice job answering the queries and comment. There are however a few additional comments and follow-up concerns based on the author’s responses.

Page 8, line 162: If there is inaccuracy in the ID numbers received from the Register service and those controls had to be excluded, that is a valid reason why there are fewer than 10 controls for each case. This also justifies why it would be important to analyze these groups in clusters since there are missing patients from some of the matched sets. It is unlikely that this would markedly influence the results, but this should however be disclosed and explained to the reader in the Results and Discussion.

Figure 1b: If possible, it would be preferable to combine both cases and controls into one flow diagram in order for the reader to appreciate where the cases and matched controls originated from. From the new diagram it appears that the exclusion criteria of no chromosomal aberrations was applied to the cases, but not to the control patients. However in the text (Page 7, line 143) it seems that this exclusion criteria was made. It would be helpful to include this in the flow diagram so the readers can see that the same exclusions were made to both cohorts.

Table 1: Thank you for the clarification. It makes sense that if the authors cannot differentiate the patients, it would be impossible to give the patients characteristics for only the 485 included individuals. However, this should be disclosed in a footnote for this demographic table, that the table includes data for 38 patients who were ultimately excluded from analysis due to chromosomal anomalies. Otherwise it would be confusing to the reader why the numbers do not match up.

Page 9, line 172 and Table 1: It should also be mentioned that the additional surgeries did not necessarily occur during the study period, and could have occurred at any time during the follow-up period which was between 0 and 36 years of age. Otherwise it looks like your exposure cohort had a multiple surgery rate of nearly 50% at under 1 year of age.

Table 2: It should also be disclosed that this table of procedures and operating room times included procedures for 38 patients who were excluded from analysis as a footnote in this table.

Page 7, line 140: The authors mention that they matched on age, sex, and gestational week. In their response to a reviewer comment, they mention that they also match on year of birth to ensure that children were born in the same era. If year of birth is an additional matching criteria, it would be helpful to mention this in the text. Also if the authors have data on these variables for the cases and controls including the year of birth, it may be helpful to add this data to the Table 1 patient characteristics table.

Ideally it would be helpful to display the characteristics for the controls in a patient characteristics table. Is that possible, or do the authors not have any data for the controls other than ID numbers to match to their cases? Also It seems that authors cannot identify which cases got matched since they were unable to display the characteristics of which of the 523 cases were chosen and which were excluded. I believe the Swedish Birth Register data has been widely used and is likely to be reliable. However, these aspects of the study should likely be mentioned if this is the case.

Page 7, line 155: If a mixed ordinal regression was used to take into account matched sets and missing controls in the primary analysis, shouldn’t a similar analysis be used in the ADHD analysis instead of Fisher’s exact tests since the title of the study is now the evaluation of ADHD?

7. PLOS authors have the option to publish the peer review history of their article (what does this mean?). If published, this will include your full peer review and any attached files.

Reviewer #2: No

---

## [Author Response · Author response to Decision Letter 1]

25 Sep 2020

Review Comments to the Author

Reviewer #2: The authors did a nice job answering the queries and comment. There are however a few additional comments and follow-up concerns based on the author’s responses.

Page 8, line 162: If there is inaccuracy in the ID numbers received from the Register service and those controls had to be excluded, that is a valid reason why there are fewer than 10 controls for each case. This also justifies why it would be important to analyze these groups in clusters since there are missing patients from some of the matched sets. It is unlikely that this would markedly influence the results, but this should however be disclosed and explained to the reader in the Results and Discussion.

Response: We have added this extra information plus an explaining table (Table 1) in Results page 8 lines 164-166 and in the discussion section page 16 lines 319-321.

Figure 1b: If possible, it would be preferable to combine both cases and controls into one flow diagram in order for the reader to appreciate where the cases and matched controls originated from. From the new diagram it appears that the exclusion criteria of no chromosomal aberrations was applied to the cases, but not to the control patients. However in the text (Page 7, line 143) it seems that this exclusion criteria was made. It would be helpful to include this in the flow diagram so the readers can see that the same exclusions were made to both cohorts.

Response: Thank you for this input. We have now made a new flow diagram which replaces Fig 1a and 1b. 

Table 1: Thank you for the clarification. It makes sense that if the authors cannot differentiate the patients, it would be impossible to give the patients characteristics for only the985 included individuals. However, this should be disclosed in a footnote for this demographic table, that the table includes data for 38 patients who were ultimately excluded from analysis due to chromosomal anomalies. Otherwise it would be confusing to the reader why the numbers do not match up.

Response: The information that it includes individuals that later were excluded for chromosomal aberration is already in the table. However, we now added the number, 38, to the table footnote to make it clearer to the reader plus added an extra asterix to “highlight the information”.

Page 9, line 172 and Table 1: It should also be mentioned that the additional surgeries did not necessarily occur during the study period, and could have occurred at any time during the follow-up period which was between 0 and 36 years of age. Otherwise it looks like your exposure cohort had a multiple surgery rate of nearly 50% at under 1 year of age.

Response: We have added this information to the table as a footnote as well as on page 9 line 179.

Table 2: It should also be disclosed that this table of procedures and operating room times included procedures for 38 patients who were excluded from analysis as a footnote in this table.

Response: We have added this information to the table as a footnote.

Page 7, line 140: The authors mention that they matched on age, sex, and gestational week. In their response to a reviewer comment, they mention that they also match on year of birth to ensure that children were born in the same era. If year of birth is an additional matching criteria, it would be helpful to mention this in the text. Also if the authors have data on these variables for the cases and controls including the year of birth, it may be helpful to add this data to the Table 1 patient characteristics table.

Ideally it would be helpful to display the characteristics for the controls in a patient characteristics table. Is that possible, or do the authors not have any data for the controls other than ID numbers to match to their cases? Also It seems that authors cannot identify which cases got matched since they were unable to display the characteristics of which of the 523 cases were chosen and which were excluded. I believe the Swedish Birth Register data has been widely used and is likely to be reliable. However, these aspects of the study should likely be mentioned if this is the case.

Response: The year of birth is not an additional matching criteria, it is the same criteria as the criteria for matching on age. We have also added information on median age and median gestational week of the cases and control group to page 8 lines 167-169.

Page 7, line 155: If a mixed ordinal regression was used to take into account matched sets and missing controls in the primary analysis, shouldn’t a similar analysis be used in the ADHD analysis instead of Fisher’s exact tests since the title of the study is now the evaluation of ADHD?

Response: In theory it would be possible to analyze ADHD using a mixed logistic regression, however limitations in the data causes issues with convergence when estimating the model. The issue is caused due to the few number of ADHD diagnoses in the group of cases.

---

## [Decision Letter · Decision Letter 2]

6 Oct 2020

Attention deficit hyperactivity disorder and educational level in adolescent and adult individuals after anesthesia and abdominal surgery during infancy

PONE-D-20-10556R2

Dear Dr. Arana Håkanson,

We’re pleased to inform you that your manuscript has been judged scientifically suitable for publication and will be formally accepted for publication once it meets all outstanding technical requirements.

Kind regards,

Antonio Palazón-Bru, PhD

Academic Editor

PLOS ONE

Additional Editor Comments (optional):

Reviewers' comments:

Reviewer's Responses to Questions

**Comments to the Author**

1. If the authors have adequately addressed your comments raised in a previous round of review and you feel that this manuscript is now acceptable for publication, you may indicate that here to bypass the “Comments to the Author” section, enter your conflict of interest statement in the “Confidential to Editor” section, and submit your "Accept" recommendation.

Reviewer #2: All comments have been addressed

2. Is the manuscript technically sound, and do the data support the conclusions?

Reviewer #2: Yes

3. Has the statistical analysis been performed appropriately and rigorously? 

Reviewer #2: Yes

4. Have the authors made all data underlying the findings in their manuscript fully available?

Reviewer #2: No

5. Is the manuscript presented in an intelligible fashion and written in standard English?

Reviewer #2: Yes

6. Review Comments to the Author

Reviewer #2: (No Response)

7. PLOS authors have the option to publish the peer review history of their article (what does this mean?). If published, this will include your full peer review and any attached files.

Reviewer #2: No

---

## [Editor Report · Acceptance letter]

12 Oct 2020

PONE-D-20-10556R2 

Attention deficit hyperactivity disorder and educational level in adolescent and adult individuals after anesthesia and abdominal surgery during infancy 

Dear Dr. Arana Håkanson:

I'm pleased to inform you that your manuscript has been deemed suitable for publication in PLOS ONE. Congratulations! Your manuscript is now with our production department. 

Kind regards, 

on behalf of

Dr. Antonio Palazón-Bru 

Academic Editor

PLOS ONE